# Dual Sensory Impairment: The Correlation between Age Related Macular Degeneration and Sensorineural Hearing Loss

**DOI:** 10.3390/medicina58020291

**Published:** 2022-02-14

**Authors:** Marina Istrate, Mihai Hasbei-Popa, Daniela Adriana Iliescu, Ana Cristina Ghiță, Brigitha Vlaicu, Mihai Aurelian Ghiță

**Affiliations:** 1Department of Doctoral School, “Victor Babes” University of Medicine and Pharmacy, 300041 Timisoara, Romania; 2Department of Ophthalmology, “Infosan” Ophthalmology Clinic, 010538 Bucharest, Romania; 3Department of Medicine, “Iuliu Hatieganu” University of Medicine and Pharmacy, 400000 Cluj-Napoca, Romania; 4Department of Physiology, ”Carol Davila” University of Medicine and Pharmacy, 050474 Bucharest, Romania; 5Department of Ophtalmology, ”Ocularcare” Eye Clinic, 012244 Bucharest, Romania; 6Department of Hygiene, ”Victor Babes” University of Medicine and Pharmacy, 300041 Timisoara, Romania

**Keywords:** age-related macular degeneration, melanin, retinal pigment epithelium, cochlea, sensorineural hearing loss

## Abstract

The pathogeneses of age-related macular degeneration (AMD) and age-related hearing impairment are not yet fully understood. If AMD and age-related hearing impairment are correlated, the cause of both may be a result of a common vulnerability. The aim of this study was to assess the interrelation between age-related macular degeneration and age-related hearing loss. Material and methods: In our case-control analysis, the hearing conditions of 40 subjects with AMD were compared with 40 age-matched healthy controls. In all patients, retinal changes were certified by clinical examinations, optical coherence tomography (OCT), and fluorescein angiography (FA). All subjects were inspected with pure tone audiometry (PTA), impedance audiometry, and speech audiometry. Results: A significant correlation (*p* < 0.001) was identified between age-related macular degeneration and age-related hearing impairment. The predominant hearing impairment in this case was sensorineural (SNHL). Of the patients diagnosed with AMD, SNHL was found in 88.89% of those with exudative macular degeneration and in 67.74% of those with atrophic macular degeneration. In contrast, we found that a significant proportion of the control group had normal hearing. Conclusion: One possible explanation for the association between retinal and cochlear impairment may be due to a melanin disorder.

## 1. Introduction

Macular degeneration is a multifactorial disorder of the retina and a major cause of central vision loss in the elderly. The prevalence of age-related macular degeneration (AMD) varies in different racial and ethnic groups around the world and rises with each decade after the age of 50. The etiology of AMD is not well-established; however, the main role has been assigned to retinal pigment epithelium (RPE) illness. Melanin in the RPE protects photoreceptor cells against light-related toxicity and plays a crucial role in the pathogenesis of AMD [1,2]. Melanin belongs to a family of biomolecules that determines the pigmentation of various tissues. The melanic pigments (eumelanin, pheomelanin and neuromelanin) are the result of intricate biochemical pathways starting from the aromatic amino acid tyrosine. Melanin has an important photoprotective function and is involved in the neutralizing reaction of free radicals and reactive oxygen species [3]. On the other hand, regarding the auditory system, the stria vascularis of the cochlea is the main structure containing melanocytes. Small melanin density makes the cochlea more sensitive to oxidative damage and lipofuscin aggregation [4,5]. The present study aimed to analyse the association between macular and cochlear damage and to provide a summary report about the contribution of melanin to retinal and cochlear metabolic processes.

## 2. Materials and Methods

### 2.1. Study Subjects

The study protocol was approved by the ethics committee of “Victor Babes” University of Medicine and Pharmacy, Timisoara, Romania (Approval No. 05/2019, Approval date 10 January 2019). All procedures performed in this study involving human participants were in accordance with the ethical standards of the institutional and national research committee and with the Helsinki Declaration and its later amendments or comparable ethical standards. Informed consent was obtained from all individual participants included in the study.

In our case-control study, the hearing conditions of 40 patients with age-related macular degeneration (AMD) were compared with 40 age-matched healthy controls. The studied population was referred to Ocularcare Eye Clinic during one year from March 2019 to March 2020. In all patients, retinal changes were confirmed by posterior pole eye exams, fluorescein angiography, and optical coherence tomography.

### 2.2. Quantification of Lesion Morphology

The diagnosis of retinal lesions was performed by a retinal specialist in every case. Due to the variety of retinal lesions, optical coherence tomography (OCT) was followed by fluorescein angiography (FA). Nine types of lesion components were established. Drusen was defined as a white-yellow retinal lesion located at the interface between the Bruch membrane and retinal pigment epithelium. On the fluorescein angiography, multiple hyperfluorescent spots of small size and round shape were detected with poorly defined contours and a tendency to confluence. Geographic atrophy (GA) was recognized by some patterns, including drusen regression and progression of the atrophy in areas with pigmentary changes. FA was detected in atrophic areas with a window defect and with good visibility of the choroidal vessels. Choroidal neovascularization (CNV) was clarified by a blood vessel complex that expands through the Bruch membrane from the choriocapillaris into the sub-RPE (type 1) or subretinal space (type 2). Classic CNV was defined as a well-defined laced pattern during early transit with subsequent leakage and late staining of fibrous tissue. Occult CNV was described as an elevated, irregular area of hyperfluorescence in the middle phase and with persistent hyperfluorescence in the late phase of FA. Retinal pigment epithelial detachment (PED) from the inner collagenous stratum of Bruch membrane was identified as a round-ovalar lesion with smooth edges that showed early hyperfluorescence and leakage. Four types of PED were observed: serous, drusenoid, fibrovascular, and haemorrhagic. Serous PED was identified by a well-circumscribed area of hyperfluorescent pooling. Drusenoid PED was identified by an early diffuse hypofluorescence and irregular late staining. Fibrovascular PED showed irregular, stippled hyperfluorescence, leakage, and late staining on FA. Haemorrhagic PED was identified by an abundant masking of background fluorescence but with visibility of the overlying vessels. A disciform scar was recognized by a prompt staining and a strong hyperfluorescence without leakage.

### 2.3. Audiological Screening

All patients were investigated with audiological tests comprising pure tone audiometry (PTA), speech audiometry, tympanometry, the measurement of the Stapedius reflex, and otomicroscopic examination. Audiometry was effectuated by using an AMPLIVOX Model 240 Portable Diagnostic Audiometer by an otologist. The cases of congenital hearing damage and any auditory conditions related to ototoxic drugs or noise exposure were eliminated from the study. The examination of hearing-impaired subjects and audiometry analysis were performed in close collaboration with an otolaryngologist.

#### 2.3.1. Pure Tone Audiometry

The measurements of air and bone pure-tone thresholds were obtained to provide information regarding hearing sensitivity and the type and severity of hearing impairment.

Pure-tone thresholds were evaluated in concordance with the International Organization for Standardization (ISO). Pure tones for the frequencies 125, 250, 500, 1000, 2000, 4000, and 8000 Hertz (Hz) were obtained for the air conduction test, and the frequencies 250, 500, 1000, 2000, 4000, and 8000 Hz were obtained for the bone conduction test (Table 1).

The audiological screening was performed by speech audiometry, tympanometry, the measurement of the Stapedius reflex, and otomicroscopic examination, in accordance with international reference standards.

#### 2.3.2. Type of Hearing Impairment

The type of hearing impairment was classified as normal hearing, conductive hearing loss, sensorineural hearing loss, and mixed hearing loss in accordance with the World Health Organization (WHO). Therefore, the normal hearing range was <20 dB HL. A conductive hearing loss was defined as air-bone gaps ≥15 dB at one frequency or ≥10 dB at two frequencies. A sensorineural hearing loss was defined as a hearing threshold ≥20 dB HL with an air-bone gap <15 dB at one frequency or <10 dB at two frequencies. Mixed hearing loss was defined as a combination of both conductive and sensorineural hearing loss.

### 2.4. Statistical Analysis

The statistical analysis was performed using SPSS software version 22.0. Pearson’s correlation analysis was used, and a *p* value of less than 0.05 was taken as statistically significant. We used statistical tests such as the Kruskal–Wallis test, the Mann–Whitney test, ANOVA, *t*-test, chi-squared test, and the Wilcoxon signed-rank test. The distribution of variables was validated using the Shapiro–Wilk test. The Kolmogorov–Smirnov test was used to verify the assumption of normality.

## 3. Results

We detected that hearing thresholds at frequencies of 125, 250, 4000, and 8000 Hz were statistically higher in the cases shown (Table 2). Table 3 shows the mean of hearing threshold levels in cases based on *t*-test analysis. The average degree of hearing loss in varied subtypes of AMD (drusen, geographic atrophy, PED, scar, CNV) was notably different from the control group (Table 4).

In this case-control study we remark that AMD was found to be statistically related to hearing impairment (*p* < 0.001), especially sensorineural hearing impairment (SHI). A significantly higher percentage of SNHL was noticed in AMD patients compared with the control group (Figure 1). From the patients diagnosed with AMD, SNHL was found in 88.89% of those with exudative macular degeneration and in 67.74% of those with atrophic macular degeneration. In contrast, we found that a significant proportion of the control group had normal hearing (80%). Comparable mixed and conductive hearing damage was observed in cases of atrophic AMD and in control subjects. In the control group, just 5% had SNHL. On the other hand, in cases of atrophic macular degeneration, normal hearing was found only in 9.677%. No cases with exudative macular degeneration had normal hearing or conductive hearing loss.

## 4. Discussion

The present case-control study revealed that AMD was significantly associated with sensorineural hearing impairment. One possible hypothesis for the correlation between retinal and cochlear damage may be the melanin disorder present in RPE and in the inner ear.

Melanin is located in many tissues and has a lot of different properties throughout the body. Most of these biochemical and physical properties are related to a defense mechanism against oxidative stress injury. Besides photoprotective functions, melanin defends the cells from cytotoxic repercussions caused by inflammatory processes and acts like a shield against environmental factors [6].

### 4.1. Synthesis of the Melanic Pigments

Melanin is synthetized in a group of cells known as melanocytes. The tyrosine and its hydroxylated DOPA product are key for melanic pigments synthesis. There are three types of melanin: pheomelanin, eumelanin, and neuromelanin. Pheomelanin is a yellowish-reddish pigment which is obtained from the oxidative polymerization of the 5-S-cysteinyldopa. Eumelanin is synthetized through the oxidative polymerization of 5,6-dihydroxyindole. There are brown and black eumelanins. Neuromelanin is a dark pigment biosynthesized from levodopa [7].

### 4.2. Melanin Functions

Melanin blocks infrared light and UV radiation. This pigment is efficacious to neutralize free radicals and reactive oxygen species, and it reduces cytotoxic lipid peroxidation. Melanin has the ability to chelate metal ions and bind to a multitude of lipophilic compounds, xenobiotics, and organic molecules. Last, but not least, melanin decreases light toxicity and protects tissues from photodamage [6].

### 4.3. Role of Ocular Melanin

The human eye consists of layers of pigmented tissues that include melanin. Melanin pigment is naturally found in the uvea and RPE. In the anterior segment of the eye, melanocytes are located in the stroma of the iris and ciliary body. In the choroid layer, melanocytes are concentrated in suprachoroidal space and choroidal stroma. The melanin found in the cytoplasm of RPE cells has an important photoprotective role. The role of melanin in ocular tissues is not yet fully understood. Melanin is an important factor for eye protection against ophthalmic disorders, such as age-related macular degeneration (AMD) and uveal melanoma. Uveal melanocytes in eyes with dark-colored irises have more eumelanin compared with light-colored eyes and thus can protect the eye from light toxicity. Because eumelanin is less photoreactive than pheomelanin, the elevated level of eumelanin in dark-colored eyes is associated with a smaller frequency of uveal melanoma and age-related macular degeneration, as epidemiological studies suggest. Furthermore, choroidal melanin is an antioxidant, it neutralizes reactive oxygen species injury, and it defends the retina from oxidative stress. The distribution of the RPE melanin is also very important in considering the toxicity of drugs related to it. In contrast to the choroidal layer, RPE melanin has a zonal allotment, with decreasing levels from the equator but with a peak concentration at the macula lutea. In the macular region the cytoarchitectonics of the RPE cells is narrower. Therefore, melanin concentration is greater in the macular area where light is focused.

The protective functions of melanin on the eye tissues seem to be performed by biochemical and physical mechanisms. The pigment has antioxidant and photo-screen effects. The photo-screening effect prevails in the anterior pole of the eyeball, which is vulnerable to sunlight and UV radiation. The posterior pole of the eye is exposed to a limited quantity of light and ultraviolet waves. Therefore, antioxidant properties are the main way of protection [8,9].

### 4.4. Distribution and Role of Inner Ear Melanin

In the human ear, the melanin pigment is mostly present in the cochlea, in the vestibular organ, and in the endolymphatic sac. Melanin-containing cells of the cochlea are located in the stria vascularis and the modiolus. Melanin is abundant in the intermediate layer of the three-layered vascularized epithelial tissue. The literature suggests that all phases of local melanogenesis occur in the intermediate cells. In the modiolus, melanocytes are distributed in the perineural and perivascular spaces.

In the vestibular organ, melanocytes are distributed in two sectors: (1) in the utricle and the ampulla and (2) in the saccule. The melanocytes of the ampulla and utricle are found subepithelially, below the basement membrane among the connective tissue cells. In the saccule, melanin pigment is found in the membranous wall. In the endolymphatic sac, the melanocytes are located in properly vascularized areas, bordering with the cells responsible for the support of the endolymphatic ionic content. Melanin protects the inner ear against oxidative damage. Low levels of melanin make the cochlea vulnerable to oxidative stress and lipofuscin aggregation [10,11,12].

Through aging, the quantity of melanin in RPE cells and the cochlea decreases, and lipofuscin depositions rise. Small melanin concentrations seem to be one of the most important factors leading to AMD and SNHL development.

Several studies emphasize the correlation between age-related hearing impairment and age-related macular degeneration. The study conducted by Ghasemi et al. denoted a clear association between simultaneous macular and cochlear impairment. Bozkurt et al. underlined that age-related hearing loss is more common in patients with age-related macular degeneration.

## 5. Conclusions

Our study denotes a significant association between age-related macular degeneration and sensorineural hearing loss. We would like to emphasize the possibility of a common melanin impairment in RPE cells and the inner ear. Therefore, melanin degradation may account for a major cause for simultaneous hearing and visual damage in the elderly. Further studies are necessary to conclude a causal association between age-related macular degeneration and sensorineural hearing loss.

## Figures and Tables

**Figure 1 medicina-58-00291-f001:**
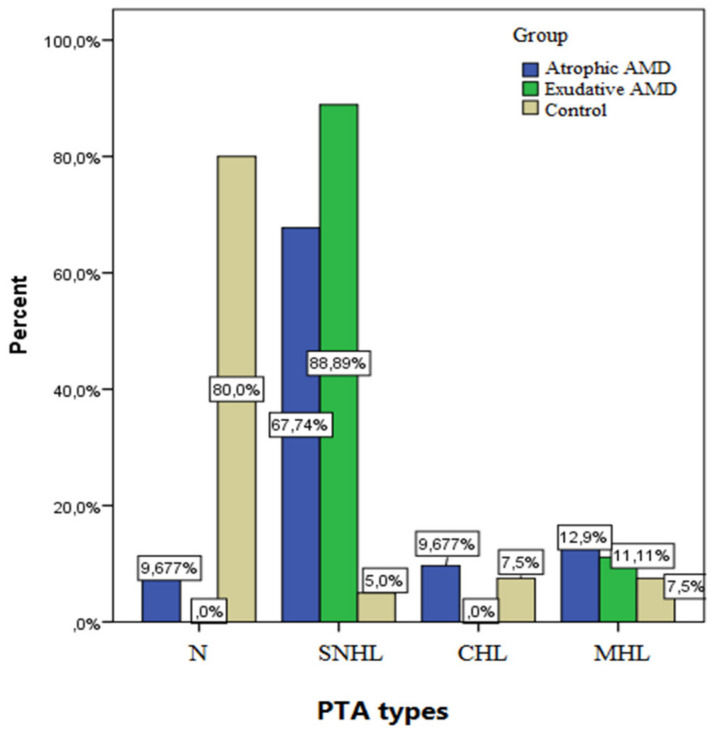
Frequency of PTA types in cases and control group. PTA, pure tone audiometry; SNHL, sensorineural hearing loss; CHL, conductive hearing loss, MHL, mixed hearing loss; AMD, age-related macular degeneration.

**Table 1 medicina-58-00291-t001:** Maximum hearing levels provided at each frequency.

Frequency, Hz	Air Conduction, dBHL	Bone Conduction, dBHL
125	80	-
250	100	45
500	115	60
1000	120	70
2000	120	70
4000	115	70
8000	100	40

Hz, hertz; dbHL, decibels hearing level.

**Table 2 medicina-58-00291-t002:** Mean of hearing thresholds levels at various frequencies decibel in cases and control group.

Frequency (Hz)	Mean (dB)	SD	*p*
125 Case	19.48	15.64	<0.001
Control	8.12	6.12	
250 Case	20.2	15.3	0.001
Control	7.40	5.9	
1000 Case	19.5	18.2	0.004
Control	14.8	14.1	
2000 Case	24.2	22.2	0.017
Control	18.7	7.5	
4000 Case	36.4	24.6	<0.001
Control	26.8	20.3	
8000 Case	48.2	26.2	<0.001
Control	30.8	25.9	

Hz, Hertz; dB, decibels; SD, standard deviation.

**Table 3 medicina-58-00291-t003:** Mean of hearing thresholds levels in exudative and nonexudative age-related macular degeneration.

Types of AMD	Frequency (Hz)	Mean	SD	*p*
Nonexudative	125	17.37	12.9	<0.001
Exudative		31.14	21.1	
Nonexudative	250	12.94	11.8	0.004
Exudative		23.26	20.3	
Nonexudative	1000	16.03	15.23	<0.001
Exudative		29.84	19.4	
Nonexudative	2000	20.31	19.89	0.013
Exudative		35.21	25.16	
Nonexudative	4000	38.55	25.32	0.006
Exudative		49.23	27.44	
Nonexudative	8000	21.45	14.92	<0.001
Exudative		6.48	4.23	

AMD, age-related macular degeneration; Hz, hertz; SD, standard deviation.

**Table 4 medicina-58-00291-t004:** Mean of hearing thresholds levels in different angiographic patterns in cases and control group.

Agiographic Pattern (*n*)	Frequency 125	Frequency 250	Frequency 1000	Frequency 2000	Frequency 4000	Frequency 8000
Drusen (2)						
Mean	17.45	12.23	18.13	24.13	39.16	48.72
SD	2.36	1.45	2.01	2.12	2.41	1.98
Control (40)						
Mean	8.12	7.40	14.8	18.7	26.8	30.8
SD	6.12	5.9	14.1	17.5	20.3	25.9
*p*	<0.001					
GA (7)						
Mean	14.85	10.02	13.66	13.35	28.19	35.42
SD	1.89	7.55	7.82	1.14	2.13	1.15
Control (40)						
Mean	8.12	7.40	14.8	18.7	26.8	30.8
SD	6.12	5.9	14.1	17.5	20.3	25.9
*p*	<0.001					
Occult CNV (9)						
Mean	15.68	15.78	23.16	14.65	29.31	36.43
SD	2.13	1.23	2.29	2.14	1.88	2.24
Control (40)						
Mean	8.12	7.40	14.8	18.7	26.8	30.8
SD	6.12	5.9	14.1	17.5	20.3	25.9
*p*	0.02					
Classic CNV (5)						
Mean	36.12	26.14	32.24	41.71	55.13	58.63
SD	2.16	2.12	1.86	2.01	3.12	2.19
Control (40)						
Mean	8.12	7.40	14.8	18.7	26.8	30.8
SD	6.12	5.9	14.1	17.5	20.3	25.9
*p*	0.012					
Serous PED (2)						
Mean	17.31	16.23	15.96	24.09	41.12	45.61
SD	1.23	1.34	1.89	2.21	2.36	2.15
Control (40)						
Mean	8.12	7.40	14.8	18.7	26.8	30.8
SD	6.12	5.9	14.1	17.5	20.3	25.9
*p*	0.06					
Fibrovascular PED (7)						
Mean	36.24	27.81	36.12	41.86	54.23	38.23
SD	2.46	1.87	2.13	2.11	2.53	3.12
Control (40)						
Mean	8.12	7.40	14.8	18.7	26.8	30.8
SD	6.12	5.9	14.1	17.5	20.3	25.9
*p*	<0.001					
Drusenoid PED (1)						
Mean	16.23	14.16	25.66	34.12	29.98	32.66
SD	2.16	3.56	6.45	6.89	5.14	7.69
Control (40)						
Mean	8.12	7.40	14.8	18.7	26.8	30.8
SD	6.12	5.9	14.1	17.5	20.3	25.9
*p*	0.035					
Haemorrhagic PED (5)						
Mean	29.45	26.55	27.00	35.23	41.44	36.78
SD	1.22	1.14	2.16	1.17	1.11	2.19
Control (40)						
Mean	8.12	7.40	14.8	18.7	26.8	30.8
SD	6.12	5.9	14.1	17.5	20.3	25.9
*p*	<0.001					
Scar (2)						
Mean	2.47	27.54	28.19	36.25	39.82	41.22
SD	1.36	2.14	1.16	2.18	2.13	2.27
Control (40)						
Mean	8.12	7.40	14.8	18.7	26.8	30.8
SD	6.12	5.9	14.1	17.5	20.3	25.9
*p*	<0.001					

GA, geographic atrophy; CNV, choroidal neovascularization; PED, pigment epithelial detachment; SD standard deviation.

## Data Availability

Not applicable.

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
