# Peer review of "Dual Sensory Impairment: The Correlation between Age Related Macular Degeneration and Sensorineural Hearing Loss"

_medicina, 2022, doi:10.3390/medicina58020291_

Round 1
Reviewer 1 Report
Comments reviewer n°1. MS 1539383-
Dual Sensory Impairment: The Correlation Between Age Related Macular Degeneration and Sensorineural Hearing Loss
I have read with interest the article entitled “Dual Sensory Impairment: The Correlation Between Age Related Macular Degeneration and Sensorineural Hearing Loss”
The authors in this case / control study (ethical committee approbation N° 05/2019)performed in 40 patients and 40 subjects(age matched) in a tertiary Ophtalmological center and studied with PTA for hearing analysis and clinical examination, OCT and fluorescein angiography for Ophtalmological analysis conclude to a correlation between age related hearing loss and macular degenerescence (mainly for the oedematous form) and suggest a possible common point at the level of pigmentation cells for the role of melanin as a protector.
Minor remarks
-In the abstract probable typing error line 29: “ One possible explantation for…” should possibly be : « One possible explanation for.. »
-Introduction line 36 the ? AMD is mentioned for the first time in the text (although it is only and already explained in the abstract) and should be explicit also in the text ( firstly mentioned citation in the main text); Age Related Degeneration
-Material and Methods line 73: “FA was defined…” should be reformulated
Line 76”chronic CNV was defines…” should be: “chronic CNV was defined..”
Other remarks
The different locations of melanin pigments and their role in the anterior and posterior segment of the eye and in the inner ear endolymphatic sector (utricle, saccule, stria vascularis and endolymphatic sac) are well described but one could regret not to find any real breakthrough nor very recent element when compared to previous publications as in the Ghassemi paper in 2016 or Mujica-Mota MA et al (2015) in his paper “eye colour as a risk factor for acquired sensori neural hearing loss: a review. Hear Res. 2015; 320”
For the hearing impairment, a few authors have proposed the interest of Otoemissions (OAE) as Driscall C et al. Racial heritage and otoacoustic emission measures of cochlear function . Asia Pacific J Speech lang. hear 2009; 12. Or more recently Klopper M et al(2019):”The correlation between hair and eye color and contralateral suppression of otoacoustic emissions. Noise Health 2019;21 (101):155”
Such considerations about electrophysiological explorations could have been evoked in the discussion
The audiological explorations are poor and only based on the PTA. Audiometry and Hearing in noise and Speech understanding could have been interesting to report.
The reason of a higher correlation with Oedematous retinal lesions than with other lesions could have been further discussed since it seems to be the main interest or recent data brought by this work.
These results and conclusions are very similar to those mentioned in 2016 by Ghassemi H et al in J. Ophtalm Vis Res but do not bring recent arguments. There are not real new or original information and this works looks as a confirmation of previous works .
The auditory explorations are poor and in the discussion the role of skin pigmentation vs hearing loss could have been further discussed or emphasized in the light of autoimmune diseases such as Vogt-Koyanagi-Harada(VKH) syndrome ( Fatil YB et al; “VKH syndrome. J Family Med Prim Care 2020” ) which emphasizes the role of melanin.
Such correlations between melanin pigmentation and hearing loss have been reported in numerous congenial diseases (Waardenburg syndrome, Tietz syndrome..)
In a similar direction the work from Murillo-Cuesta S et al (Melanine precursors prevent premature age-related and noise induced hearing-loss in albino mice (pigment cell melanoma res. 2010) is contributive.
The role of melanin on the calcium (Ca) homeostasis is crucial for transduction of sound and endocochlear electrical Potential constitution and obtention of a compound AP.(Klopper et al)
Overall this presentation is interesting , clear and the hypothesis of a common protective role of melanin is promising and possibly convincing but needs more clues, improvements and possible further developments. This work could be considered as a first step in this direction since it is in its present form a simple confirmation of previous works and needs fore coming evolvements. The audiological exploration is possibly insufficient and shallow ( no electrophysiological exploration or data such as BERA or acoustic otoemissions. (OAE and DPOAE) which are not otherwise commented in the discussion in the light of other works)
Author Response
Dear Reviewer,
Thank you for giving us the opportunity to submit a revised draft of the manuscript “Dual Sensory Impairment: The Correlation Between Age Related Macular Degeneration and Sensorineural Hearing Loss”. We appreciate the time and effort that you and the reviewers dedicated to providing feedback on our manuscript and are grateful for the insightful comments on and valuable improvements to our paper. We have incorporated most of the suggestions made by the reviewers. Those changes are highlighted within the manuscript. The aim of this study was to correlate age related macular degeneration and age related hearing loss (this is the subject of my doctoral thesis). Unfortunately, we did not have the opportunity to investigate the patients with electrophysiological exploration or acoustic otoemissions (OAE and DPOAE). However, this limitation did not affect the overall conclusion of the study.
Thank you for your consideration sincerely,
Dr. Marina Istrate
Reviewer 2 Report
How did the authors ensure repeatability and reproducibility of both visual and audiological results?
How did the authors define wet AMD and dry AMD? Did the study include intermediate AMD?
The authors should also refer to retinal fluids (IRF, SRF) and consider when measuring ocular parameters.
Please add a patient characteristics table as it is essential to compare the control group with the studied group.
What were the inclusion and exclusion criteria? Were there any limitations? Can the authors clearly state them?
The authors should differentiate and conduct separate studies for wet AMD and Dry AMD patients and test their melanin level; only then, can they prove a significant association between the co-occurrence of wet AMD and SNHL.
The authors should discuss the visual and audiological results in the discussion and their relevancy to the current literature and how does this study contributes to the existing literature?
Author Response

(The authors gave the same response as above.)

Reviewer 3 Report
I will keep my comments short, as I have been notified that the journal has already received two complete reviews.
I very much liked the study design and data, but was far from happy with the presentation of methods and results, especially with regards to the hearing test result data.
Methods
There should be further information on methods: headphone type, are the hearing thresholds measured as air conduction or bone conduction, precision of the dB estimate, which statistical test was used, were the two ears averaged and how was this dealt with in terms of statistics? One table entry has an n of 1 and yet there is a standard deviation??!
Was this designed and executed as a prospective or retrospective study? Which patients were recruited, anyone with AMD?
Rather than “noise exposure” (which is hard to quantify) I would have asked about possible genetic or inflammatory causes of hearing loss and sudden hearing loss.
I would suggest to entirely focus on sensorineural hearing loss, as probably no one would expect any association between conductive hearing loss and AMD. By relying on bone conduction thresholds in case of an air-bone gap, this can be fairly easily be done, or one could exclude the patients with conductive hearing loss.
Discussion
A lot of the information in the discussion would be better placed in the introduction, as it summarizes current knowledge. Instead, there could be more discussion of possible common mechanisms of MD and ARHL and other associated diseases like dementia, depression, stroke etc. Or: what it would mean to affected individuals to suffer from both.
The discussion of the location and function should be improved by an expert in inner ear biology. In the text on the cochlea, you switch back and forth and I am not sure you have understood what the function of the different parts of the cochlea is. Regarding the vestibular organ, you basically describe expression everywhere. Indeed, the stria vascularis is a highly metabolically active tissue, that is likely the reason for high melanin expression. There are other types of hearing loss (e.g. Wardenburg syndrome) that are melanin disorders.
There are number of existing studies on AMD and hearing loss, e.g.
https://pubmed.ncbi.nlm.nih.gov/31336642/
https://pubmed.ncbi.nlm.nih.gov/21205373/
https://pubmed.ncbi.nlm.nih.gov/31644539/
which should be mentioned and discussed. This study still has its value in quantifying hearing thresholds in a relatively large cohort, and it backs up the large-scale population-based studies. Nonetheless, the data presented here is not as novel as it seems upon reading the manuscript.
General text
The text would benefit from improvements in English language and proofreading by an audiologist/otologist.
Figures
Hearing threshold data should be visualized in graphs similar to audiograms rather than in table format.
Figure 1 is actually a table and quite redundant with table 3
Figure 2: What was the exact measure to classify a patient as having conductive or mixed hearing loss? An average air-bone gap of x dB? I did not see this specified. Actually, what was the criterion do diagnose hearing loss at all? It would be nice to get reference values from the literature what the fraction of hearing-impaired in the respective age categories would be as an additional control group.
Table 1 please put as graph, or two columns, or in the same design as table2.
Table 2 and Table 1 could be combined?
Table 3: title: “Threshold of hearing levels” is a weird expression. Hearing threshold would be better. A graph would be better. Indeed, the controls are always the same, why are they displayed again and again?
Author Response

(The authors gave the same response as above.)

Round 2
Reviewer 2 Report
Can be accepted now